# Soil microbiota and herbivory drive the assembly of tomato plant-associated microbial communities through different mechanisms
Antonino Malacrinò [1] ✉ & Alison E. Bennett[2]

Plant-associated microbial communities are key to shaping many aspects of plant biology. In this study, we tested whether soil microbial communities and herbivory influence the bacterial community of tomato plants and whether their influence in different plant compartments is driven by microbial spillover between compartments or whether plants are involved in mediating this effect. We grew our plants in soils hosting three different microbial communities and covered (or not) the soil surface to prevent (or allow) passive microbial spillover between compartments, and we exposed them (or not) to herbivory by *Manduca sexta*. Here we show that the soil-driven effect on aboveground compartments is consistently detected regardless of soil coverage, whereas soil cover influences the herbivore-driven effect on belowground microbiota. Together, our results suggest that the soil microbiota influences aboveground plant and insect microbial communities via changes in plant metabolism and physiology or by sharing microorganisms via xylem sap. In contrast, herbivores influence the belowground plant microbiota via a combination of microbial spillover and changes in plant metabolism. These results demonstrate the important role of plants in linking aboveground and belowground microbiota, and can foster further research on soil microbiota manipulation for sustainable pest management.

Soil acts as a "seed bank" for many components of the plant microbiome, and this soil-driven effect has been reported to be stronger than other factors, such as plant genotype or herbivory[1,2]. Soil microorganisms can contribute to the plant microbiome by colonizing different plant compartments (e.g., rhizosphere, roots, leaves), both as endophytes and epiphytes. Alternatively, soil microorganisms can alter the plant physiological status (e.g., nutrition), which can, in turn, influence the microbial composition in different plant compartments (e.g., via changes in exudates or VOCs)[3]. We refer to these effects on the plant microbiome as "soil-driven." Similarly, previous studies have shown the influence of herbivory on the microbiome of different plant compartments[4–8], and this can also be due to direct changes in plant physiology or via changes indirectly induced by the herbivores (e.g., honeydew, frass). Here, we refer to this effect on the plant microbiome as "herbivory-driven." Only a few studies have focused on testing the relative strength and direction of soil- and herbivory-driven effects on the structure and diversity of plant microbiomes. For example,

Tkacz et al.[1] showed that soil has a stronger effect than plant species on shaping the plant microbiome belowground. However, little is known about the possible mechanisms that can generate soil-driven and herbivory-driven effects on microbial communities in different plant compartments.

Previous work suggests two prevailing hypotheses for how microbiomes in a plant-herbivore system might influence each other: via plant or via microbial spillover. In the first case, plants are major actors in mediating the effects on microbiomes between aboveground and belowground compartments. For instance, soil microorganisms can influence plant metabolism through their direct interaction with the host or, indirectly, by improving the availability of resources[9]. This, in turn, can alter the composition of aboveground plant tissue and directly influence the herbivore microbiome (e.g., changes in diet[10]) and indirectly (e.g., changes in leaf microbiome and consequent changes in a potential source of the herbivore microbiome[11]). In addition, soil microorganisms can become endophytes of plants and move from soil to aboveground tissue via the xylem[12]. Similarly,

[1]Dept. of Agriculture, Università degli Studi Mediterranea di Reggio Calabria, Reggio Calabria, Italy. [2]Dept. of Evolution, Ecology, and Organismal Biology, The Ohio State University, Columbus, OH, USA. ✉e-mail: antonino.malacrino@unirc.it

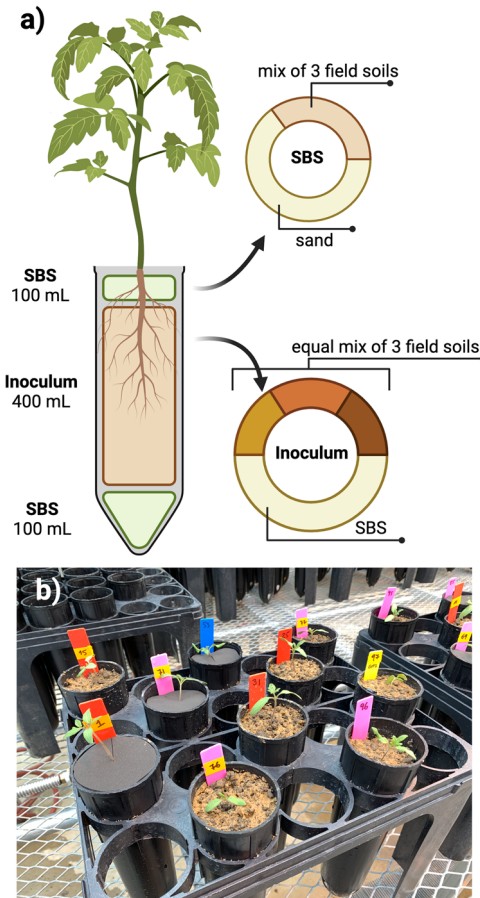

**Fig. 1 | Microcosm setup. a** Microcosm setup showing the composition of each soil layer. **b** Picture showing pots with soil covered by a neoprene disk and control plants. Created with BioRender.com.

changes in exudates or VOCs can be driven as a response to stressors (e.g., herbivory) and drive changes in the plant microbiome[3,13,14]. In contrast to the 'via plant' hypothesis, microbiomes in a plant-herbivore system might influence each other by spillover (i.e., movement of microbes between compartments). For example, herbivores can influence belowground microbial communities through microorganisms associated with frass or honeydew[15]. Similarly, soil microorganisms can reach aboveground compartments when irrigation water or rain splashes over the soil surface, and droplets containing microorganisms come into contact with leaves. Previously, we observed both soil and herbivory driven changes in microbial communities in all compartments (rhizosphere, root, leaves, herbivores), but the low overlap of microbial taxa between compartments suggested that this differentiation was driven by the compartment-specific selection of the microbial community rather than spillover between compartments[2]. Conversely, Hannula et al.[16] found that the soil microbial community did not influence the root and leaf microbiome composition but altered the herbivore-associated microbiota. Interestingly, this effect was observed when insects were feeding on potted plants but not when feeding on detached leaves. Thus, the authors suggested that caterpillars might have acquired their microbiomes directly from the soil. While the two studies differ in terms of both plant (potato vs. dandelion) and herbivore models (*Macrosiphum euphorbiae* - sap feeding vs. *Mamestra brassicae* - chewing), they suggest that the soil-driven effect on plant- and herbivore-associated microbiota might be mediated by different mechanisms.

In this study, we aimed to clarify the mechanisms underlying the reciprocal influence of microbiomes in a plant-herbivore system. Specifically, we manipulated soil microbiome composition (three soil inocula) and herbivory (presence/absence) and examined the rhizosphere, root, leaf, and

herbivore microbiota in tomato plants while covering the soil to prevent potential microbial spillover to aboveground structures from the soil (Fig. 1). Using this setup, we focused on testing: (i) whether soil microbiota composition drives changes in aboveground compartments (leaves and herbivores) via plants or via microbial spillover; and (ii) whether herbivory alters belowground microbial communities via plant or via microbial spillover. Based on previous evidence, we hypothesized that the soil- and herbivory-driven effects on plant microbiota are generated by plant-mediated mechanisms. Thus, if our hypothesis is true, we expect a soil- or herbivory-driven effect on plant microbiota regardless of the presence of soil surface cover.

## Results
### Soil inoculum, cover, and herbivory influence plant and herbivore microbiota

We tested the effects of soil inoculum, herbivory, and soil cover on the diversity of bacterial communities within each compartment (rhizosphere, roots, leaves, and herbivores) using Faith's phylogenetic diversity index as a metric (Table S1). In the rhizosphere, we found no signal driven by main factors (Table S1), but a significant interaction among the three factors ($\chi^2 = 8.46$, $df = 2$, $p = 0.01$; Table S1). Post-hoc contrasts (Fig. S1) showed no consistent patterns across treatments. In roots, soil inoculum explained 3.8% of the variance in microbial diversity, while herbivory (2.4%) and coverage (0.5%) also had minor contributions (Table S1). Post-hoc contrasts showed no consistent patterns across treatments (Fig. S2). In leaves, herbivory explained a higher proportion of the variance (9.5%, Table S1) compared to the other factors, and post-hoc contrasts within the significant herbivory by soil coverage interaction ($\chi^2 = 8.15$, $df = 2$, $p = 0.004$; Table S1; Fig. S3) showed a lower diversity in leaf samples exposed to herbivory than in the control, but only when the soil surface was covered and in agricultural and prairie soils. In herbivores, the soil inoculum explained 14.9% of the variation in microbial diversity ($\chi^2 = 7.31$, $df = 2$, $p = 0.002$; Table S1), and post-hoc contrasts showed that this effect was due to the higher phylogenetic diversity of insects feeding on plants grown on agricultural soil compared to those grown on prairie soil (Fig. 2a).

We then tested the influence of compartments (rhizosphere soil, roots, leaves, herbivores), soil inocula (agricultural, margins, prairie), herbivory (present, absent), and coverage (present, absent) on the structure of plant-associated bacterial communities using different approaches. First, we tested the influence of each factor (and their interaction) on the structure of the plant and herbivore microbiota by running separate PERMANOVA models for the rhizosphere soil, roots, leaves, and herbivores (Table 1). Soil inocula influenced the structure of the microbiota in all compartments (Fig. S4, Fig. 2b), and it was the factor explaining most variation (4.2–19.7%; Table 1, Figs. S4-S6). Herbivory explained only a minor portion of the variation (1.2–2.7%; Table 1, Fig. S5). Soil cover also explained very little variation (1.3–2.4%; Table 1; Fig. S6) and influenced the bacterial community of roots and leaves, but did not affect the rhizosphere- and herbivore-associated microbiota (Table 1).

After verifying the occurrence of a soil-driven effect on plant- and herbivore-associated microbiota, we tested whether soil cover might have influenced soil- or herbivory-driven effects. The herbivore-associated microbiota was solely influenced by soil inoculum ($F = 2.04$, $p = 0.001$; Table 1, Fig. 2b), with no effect driven by soil cover ($F = 0.024$, $p = 0.193$; Table 1). In the leaves, roots, and rhizosphere, the structure of the microbial communities was influenced by the interactions between the soil inoculum, herbivory, and coverage. Post hoc contrasts (Table S2 and S3) show that in leaves, herbivory influenced the bacterial community only in the presence of soil cover, and similarly, soil cover influenced the leaf microbiota only in the presence of herbivory. Both herbivory and soil cover influenced the root microbial communities (Tables S2 and S3), and the rhizosphere microbiota (Tables S2 and S3).

Analysis of the MNTD (Table S4) suggested that no main factor influenced the structure of bacterial communities associated with herbivores (Fig. 2c). For the three plant compartments (rhizosphere, roots,

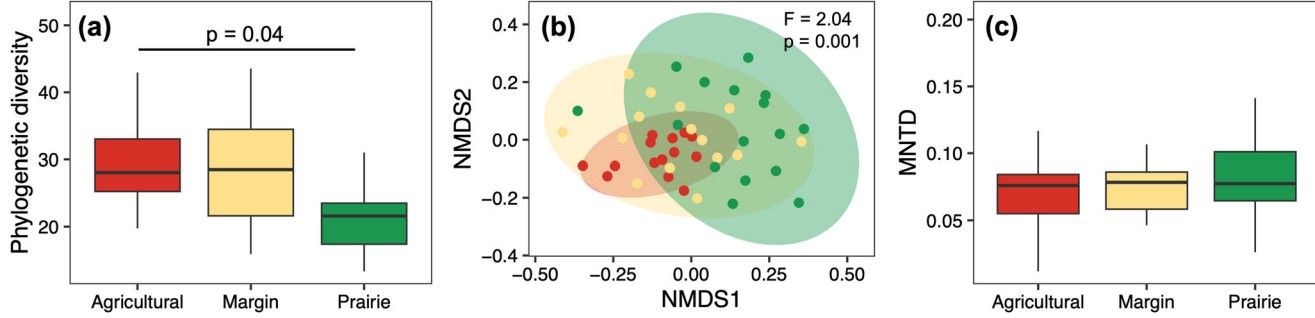

**Fig. 2 | Influence of soil inoculum on the microbial community associated with *Manduca sexta*.** Influence of soil inoculum (agricultural, margin, prairie) on *Manduca sexta* bacterial community. **a** Phylogenetic diversity, with FDR-corrected post-hoc *p*-values. **b** NMDS plots built on Unifrac distance matrix (points and 95% CI ellipses are coloured by soil inoculum, PERMANOVA results are reported on top-right corner). **c** Mean Nearest Taxon Distance (MNTD). For each group, *n* = 16.

## Table 1 | Results from PERMANOVA

| Factors | df | Rhizosphere | | | Roots | | | Leaves | | | Herbivore | | |
|---|---|---|---|---|---|---|---|---|---|---|---|---|---|
| | | $R^2$ | F | p | $R^2$ | F | p | $R^2$ | F | p | $R^2$ | F | p |
| S | 2 | 0.053 | 2.88 | 0.001 | 0.197 | 13.69 | 0.001 | 0.042 | 2.17 | 0.001 | 0.088 | 2.04 | 0.001 |
| H | 1 | 0.019 | 2.10 | 0.006 | 0.012 | 1.74 | 0.036 | 0.027 | 2.78 | 0.001 | - | - | - |
| C | 1 | 0.013 | 1.43 | 0.069 | 0.020 | 2.77 | 0.001 | 0.013 | 1.37 | 0.046 | 0.024 | 1.10 | 0.193 |
| S x H | 2 | 0.041 | 2.25 | 0.001 | 0.042 | 2.93 | 0.001 | 0.037 | 1.93 | 0.001 | - | - | - |
| S x C | 2 | 0.031 | 1.71 | 0.005 | 0.067 | 4.64 | 0.001 | 0.025 | 1.33 | 0.020 | 0.050 | 1.16 | 0.107 |
| H x C | 1 | 0.024 | 2.59 | 0.001 | 0.025 | 3.44 | 0.001 | 0.017 | 1.80 | 0.006 | - | - | - |
| S x H x C | 2 | 0.048 | 2.63 | 0.001 | 0.031 | 2.12 | 0.003 | 0.03 | 1.88 | 0.001 | - | - | - |

PERMANOVA models testing the effect of soil inoculum (S; agricultural, margin, prairie), herbivory (H; present, absent), coverage (C; present, absent), and all their interactions on the structure of plant bacterial microbiota for each compartment.

and leaves), we found a significant effect of the interaction between the three factors (soil inoculum, herbivory, and cover; Table S4); however, post-hoc contrasts did not highlight a consistent effect among the combinations of treatments (Figs. S7–S9). In addition, we also found an effect driven by soil inoculum ($\chi^2 = 10.85$, $p = 0.004$), with higher MNTD values in roots grown on prairie inoculum than in those grown on agricultural inoculum (Fig. S10). In addition, soil cover influenced the root MNTD values ($\chi^2 = 7.07$, $p = 0.007$), with higher values in samples grown without soil cover (Fig. S10).

### Soil cover shifts the relationships between microbial communities in different compartments

We found several differences in the number of shared ASVs between compartments when comparing plants grown on covered or uncovered soil surfaces (Fig. 3, Table S5). In general, we found a higher number of ASVs shared between aboveground compartments and roots when the soil surface was covered. When the soil surface was not covered, we found a higher number of ASVs shared between the aboveground compartments and rhizosphere. No differences were detected in herbivores, leaves, and ASVs shared between herbivores and leaves, and between herbivores, leaves, and the rhizosphere (Fig. 3, Table S5).

### Soil inoculum has major effects on structuring the root and herbivore microbiota

We observed three ASVs (*Flavisolibacter*, *Ramlibacter*, *Arenimonas*) that were more abundant in the rhizosphere of plants without soil cover, and two ASVs (*Pseudoflavitalea* and *Massilia*) that were more abundant in the rhizosphere of plants not exposed to herbivores (Fig. 4). In addition, soil inoculum influenced the abundance of a few ASVs in the plant rhizosphere (Fig. 5), and while the effect was inoculum-specific, three ASVs (two *Pseudomonas* and one *Chitinophaga*) were mainly associated with the

agricultural soil, whereas the other three (*Burkholderia*, unidentified Rhizobia, and *Xanthomonas*) were mainly associated with the prairie soil.

In the roots, we identified 151 ASVs whose abundance was influenced by soil coverage (78 increased, 72 decreased; Fig. 4, Supplementary Data 1). We also identified a single ASV (*Luteibacter*) influenced by herbivory (Fig. 4) and 1576 unique ASVs influenced by the soil inoculum (Fig. 5, Supplementary Data 1, Fig. S11). Among the ASVs influenced by soil inocula, 176 ASVs changed regardless of the inocula used, while 337 ASVs were influenced by specific soil treatments (133 in agricultural vs. margin, 161 in agricultural vs. prairie, 43 in margin vs. prairie).

In leaves, we observed no changes in the abundance of ASVs due to herbivory, while only one ASV (*Burkholderia*) was more abundant in plants without soil cover (Fig. 4). When comparing soil treatments, one ASV (*Pseudomonas* sp.) was more abundant in plants grown in agricultural soil (Fig. 5).

In herbivores, we found that soil cover influenced the bacterial communities by increasing the abundance of one ASV (*Pseudoflavitalea*) and decreasing the abundance of two ASVs (*Sphingomonas* and *Duganella* (Fig. 4)). Changes in the relative abundance of ASVs were unique for each soil treatment (Fig. 5).

### Soil inoculum, cover, and herbivory influence plant and herbivore biomass both directly and indirectly through changes in microbial communities

Our SEM approach (Fig. 6) expanded on the results reported above. All three factors (herbivory, soil cover, and inoculum) and the rhizosphere microbiota influenced root biomass, while shoot biomass was influenced by herbivory and soil inoculum. Soil inoculum and herbivory influenced the microbial communities of the rhizosphere microbiota. We also found that the effect driven by soil inoculum travelled up to the herbivore-associated bacterial community and this had a significant effect on insect biomass.

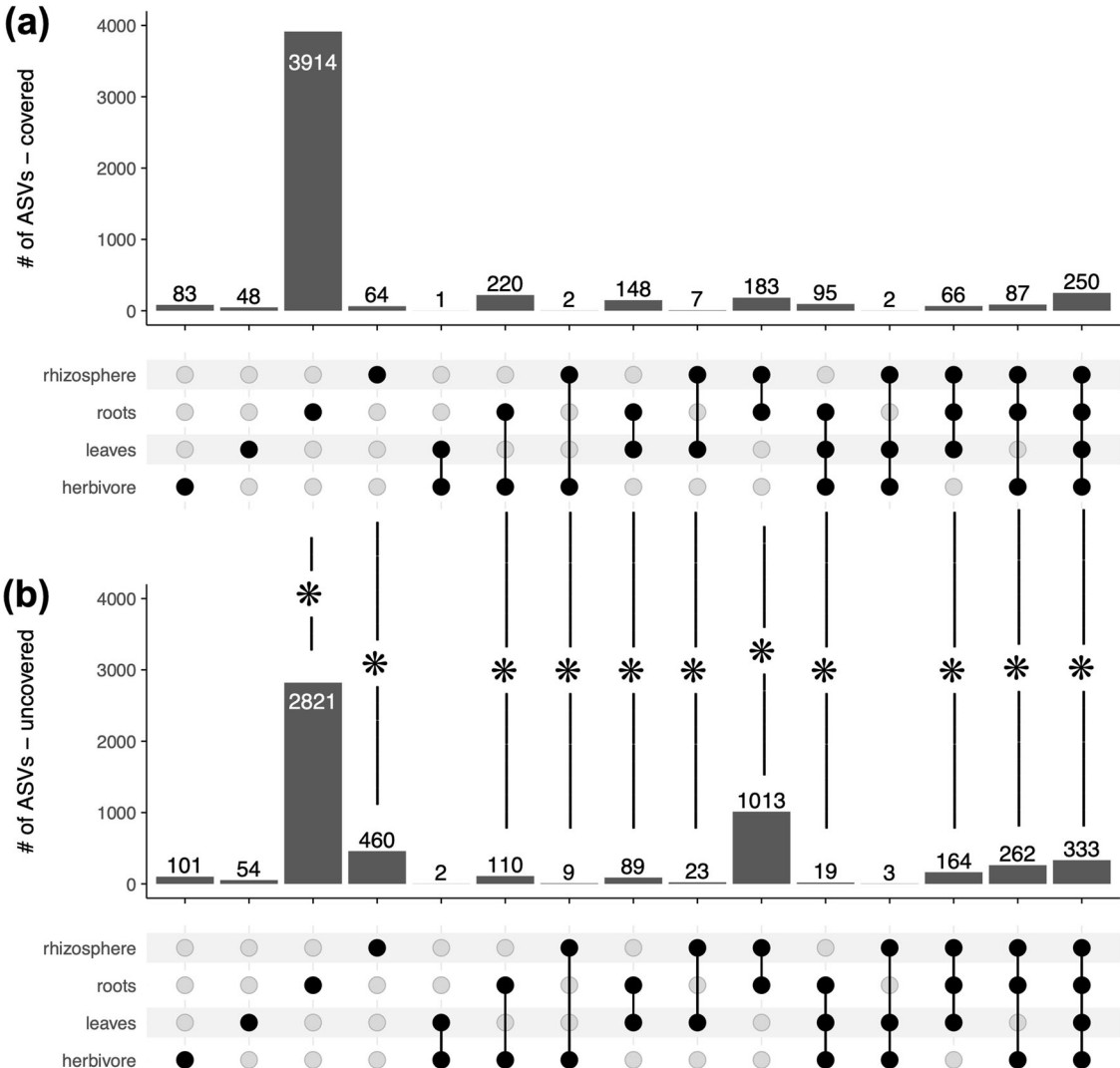

**Fig. 3 | Number of ASVs unique to a compartment or shared between compartments.** Upset plot showing the number of ASVs (Amplicon Sequence Variants) unique to a compartment or shared between compartments in plants grown on pots with (**a**) soil surface covered or (**b**) uncovered. Asterisks indicate differences between covered and uncovered plants for that intersection (see Table S5 for full details).

## Discussion

Here, we tested the influence of soil bacterial community composition and herbivory on the plant microbiota, and whether the soil- or herbivory-driven effect is mediated by microbial spillover between compartments or via host plant. We found that the soil bacterial community influences the plant microbiota composition both below- and above-ground and that the aboveground effect is mediated by the host plant and not by direct microbial transfer between the two compartments. This soil-driven effect extends to the plant- and herbivore-associated microbial communities, and negatively influenced herbivore biomass. Herbivory has a weaker effect on rhizosphere microbiota which was influenced by soil surface cover, suggesting that plants have little role on mediating the herbivory-driven effect on belowground microbiota.

Within each compartment, we found that soil inoculum was the most common factor driving the structure of bacterial communities. In particular, the soil inoculum explained a wider portion of the variance in microbiota diversity and structure in roots and herbivore microbiota, while this effect was much lower in the rhizosphere and leaves. In addition, we found a higher number of ASVs significantly affected by soil inoculum in the roots and herbivores compared to the other compartments. These results are similar to those of our previous study[2], where we observed differences

between high- and low-diversity microbial inocula in the microbiota of plants and herbivores. Here, we also found a consistent effect driven by soil inoculum on the diversity and structure of the herbivore microbiota using different metrics, and this supports our previous findings[2], despite using a different host plant (tomato vs potato) and herbivores with different feeding strategies (chewing vs sap-feeding). Previous studies have also reported a strong soil-driven effect across different plant species, including *Arabidopsis thaliana*, *Medicago truncatula*, *Pisum sativum*, *Triticum aestivum*[1], grapevine[17], and dandelion[16]. It is interesting to note that the changes driven by soil inoculum on the herbivore microbiome seem to follow a gradient of disturbance, from the most disturbed (agricultural soil) to the less disturbed (prairie soil), although further evidence is needed to test this hypothesis.

In contrast to previous studies, we investigated whether the soil-driven effect is generated by the microbial spillover between soil and the other compartments, or whether the soil microbiota exerts an effect on plants, resulting in changes in the herbivore microbiota. If the spillover hypothesis is true, leaf and herbivore microbiota should show (i) a soil-driven effect only when the soil cover is absent, and (ii) a higher proportion of shared ASVs between aboveground and belowground compartments when the soil cover is absent. Our results showed the opposite pattern. Soil inoculum influenced leaf and herbivore microbiota, regardless of soil cover. In

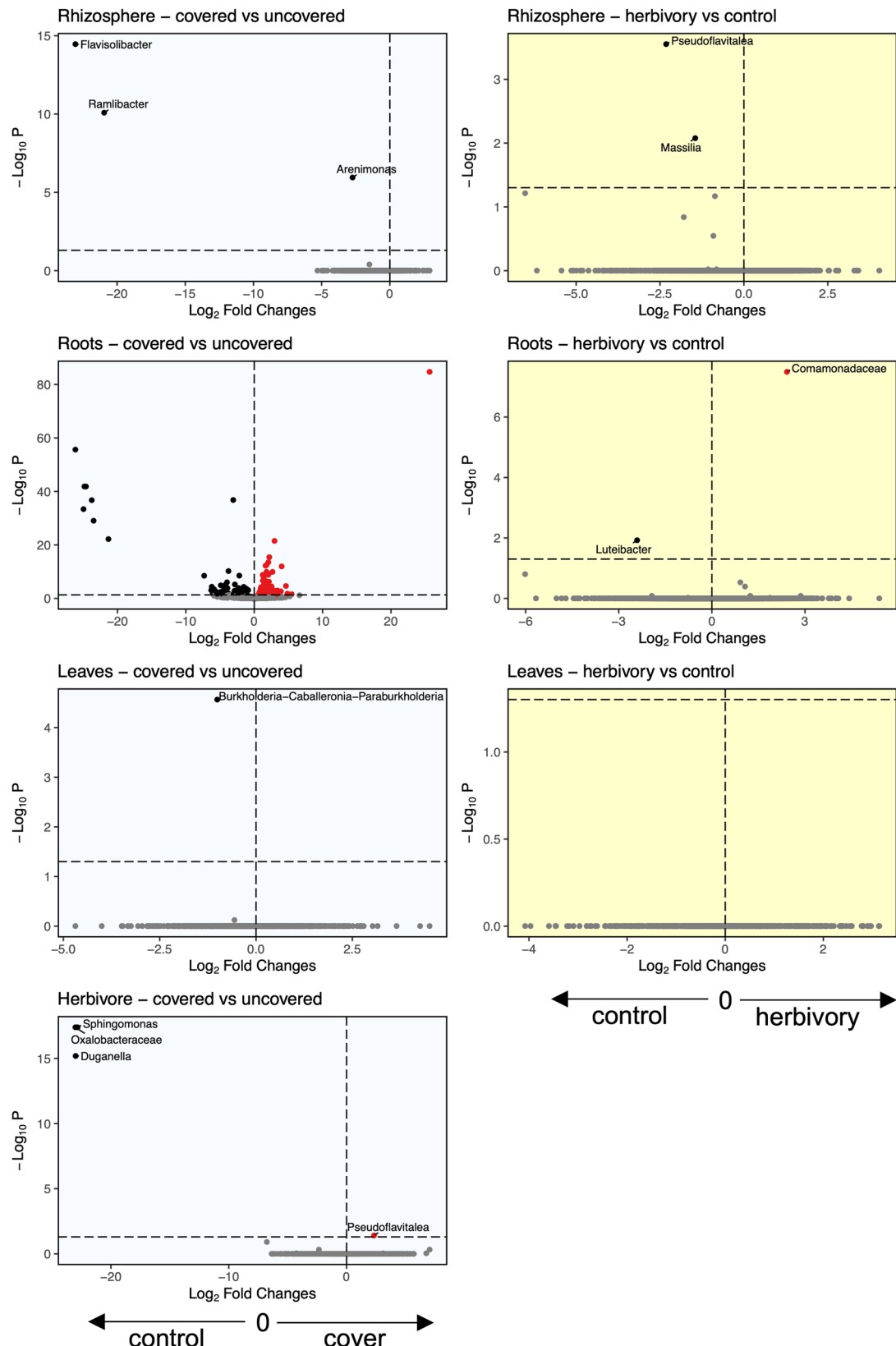

**Fig. 4 | ASVs significantly influenced by soil cover and herbivory.** Volcano plots showing the ASVs significantly influenced by soil cover (left, blue background) and herbivory (right, yellow background) for each compartment (from top to bottom: rhizosphere, roots, leaves, herbivore). For each differentially abundant ASV we report the genus name, or the family name when genus was not available. For clarity, we did not plot the genus names for the roots compartment when comparing plants with and without soil cover, and this information is available as Supplementary Data 1.

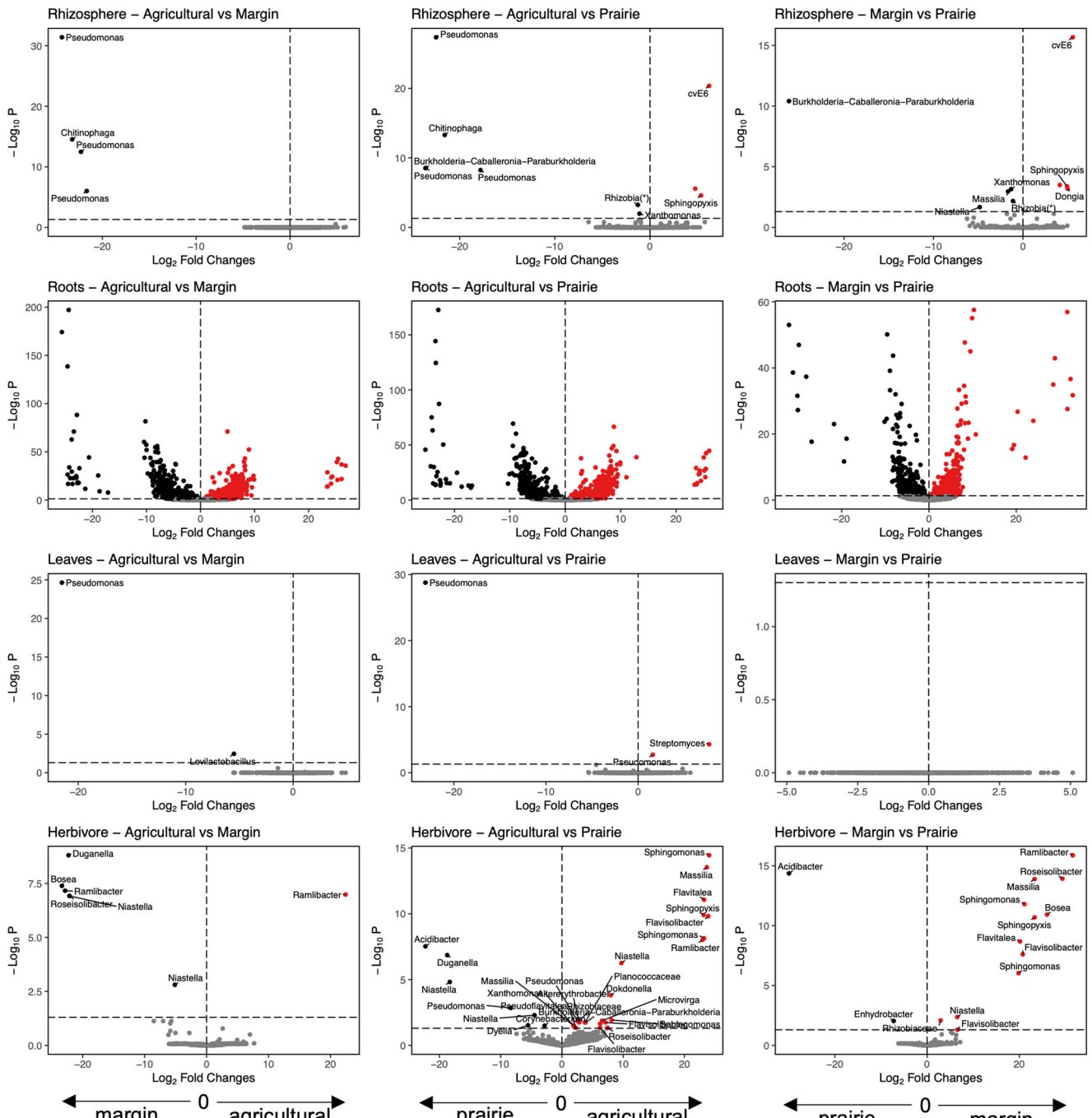

**Fig. 5 | ASVs significantly influenced by soil inoculum.** Volcano plots showing the ASVs significantly influenced by soil inoculum as pairwise contrasts between agricultural soil and field margins (left), agricultural soil and prairie soil (centre), field margins and prairie (right). Results are shown separately for each compartment (from top to bottom: rhizosphere, roots, leaves, herbivore). For clarity, we did not plot the genus names for the roots compartment, and this information is available as Supplementary Data 1. (*) For clarity, ASVs identified as *Allorhizobium-Neorhizobium-Pararhizobium-Rhizobium* according to the SILVA database are reported in the plots as *Rhizobia*.

addition, the number of ASVs shared between belowground and aboveground compartments was not influenced by soil cover, but soil coverage shifted the microbiota dynamics from a higher proportion of ASVs shared between the aboveground compartments (leaves and herbivore) with the rhizosphere to a higher proportion of ASVs shared between rhizosphere and roots. In addition, the soil-driven effect was stronger on herbivores than on leaves in multiple tests.

Thus, our results support the idea that the soil microbiota shapes plant and herbivorous microbial communities via plants. Indeed, different soil microbiota can influence plant metabolism or physiology[18,19], and this can

drive changes in leaf metabolite or physical composition[20], ultimately altering the diet of *M. sexta* and leading to changes in herbivore-associated microbiota[10,21]. Interestingly, our results showed a higher number of ASVs shared between herbivores and plant roots compared to the number of ASVs shared between herbivores and leaves and herbivores and rhizosphere. Thus, our observations suggest that roots can share components of their microbiota with herbivores. Given that soil cover had little influence on our results, meaning that microbial spillover between compartments can be excluded, an alternative mechanistic explanation to our observations might rely on the transfer of microorganisms directly from belowground to

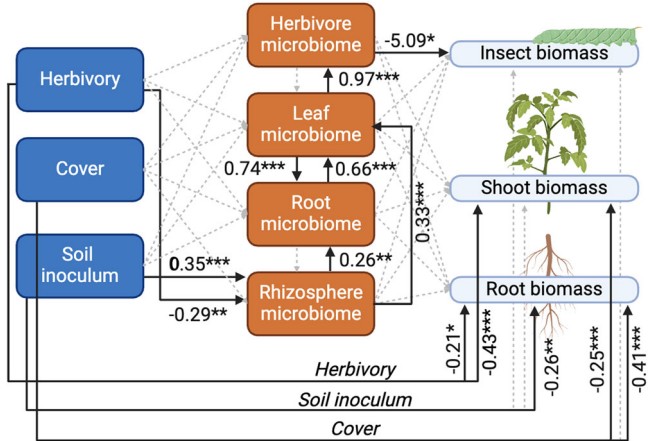

**Fig. 6 | Structural equation modelling.** Piecewise Structural Equation Modelling of the relationship between treatments (herbivory, soil cover, soil inoculum; left side), microbiota at each compartment (herbivore, leaf, root, rhizosphere; centre), and plant and insect biomass (right side). Solid lines represent only significant relationships, with standardized coefficients and significance values (***$p < 0.001$; **$p < 0.01$; *$p < 0.05$) given alongside each arrow, while dashed grey lines represent tested but not significant relationships. Created with BioRender.com.

aboveground compartments via the xylem. This hypothesis is supported by recent work on oak seedlings inoculated in a controlled gnotobiotic system, which showed that the aphid microbiota was influenced by the soil microbial community, although there was no possibility of direct contact between the two compartments[22]. However, when looking at changes in the abundance of ASVs as an effect of soil inoculum, we did not observe overlapping changes in the leaf and herbivore microbiota, which would be expected if these microorganisms were enriched via xylem sap. This incongruence might be due to the fact that we did not differentiate between leaf epiphytic and endophytic microbiota, so xylem-dwelling microorganisms might be severely under-represented in the leaf microbiota, while they might be enriched in insect guts because that niche can foster their development. Thus, future studies can help further disentangle this speculation. Interestingly, we observed that the composition of the soil microbial community can drive negative effects on insect biomass, and that this effect is mediated by changes in the plant and insect microbiota. This supports the idea that steering the soil microbiota may be an effective way to achieve sustainable pest management[23–26].

Herbivory driven changes in microbiota composition could also be plant mediated or generated by spillover between compartments, and our data contributes to parsing out support for these hypotheses as well. Other studies have also shown that herbivory drives changes in the shoot[27], root[28], and rhizosphere[5,6,29] microbiota composition. These studies posit that changes in composition are driven by changes in the metabolites, physiology, and root exudates. Humphrey and Whiteman[27] postulated that the effects of herbivory on plant-associated microbial communities could be mechanistically explained by changes in plant metabolism and physiology. In general, herbivory influenced the microbiota of the root and rhizosphere regardless of the presence of soil cover, although this effect driven by herbivory was smaller than the soil-driven effect. As discussed above, when the soil was covered, we observed a shift in the frequency of ASVs shared between the aboveground and belowground compartments. We found a higher proportion of ASVs shared between herbivore/leaves and the rhizosphere when the soil surface was not covered. This suggests that soil cover might have prevented the spillover of microorganisms directly from aboveground compartments (e.g., insect frass) to the soil and, thus, the rhizosphere. At the same time, herbivores might have exerted their influence on the root microbiota via changes in plants, as suggested by the fact that regardless of the presence of soil cover, we were still able to detect the presence of herbivores in the root and rhizosphere microbiota. Our data

suggest microbial spillover between herbivores and belowground compartments (e.g., via frass) but does not negate herbivory-driven influences on belowground microbiota via changes in plant metabolism[30]. Indeed, microbial spillover between compartments and plant-mediated changes in belowground microbiota could coexist in this system. While our study tests the effects of herbivory on the rhizosphere microbial community, it does not provide evidence on its effects on the microbiota of the bulk soil, and future studies might focus on investigating the consequences of herbivory-driven changes on the soil microbiome and their effect on the wider ecological community.

While covering soil with a neoprene disk allowed us to separate aboveground and belowground compartments from microbial spillover, it might also represent a caveat of our study. In particular, the black neoprene disks might have influenced the temperatures and/or humidity due to the black cover, similarly to black plastic mulch which increases soil temperatures and plant biomass[31]. On the other hand, the soil cover likely had a limited role in influencing the microbial communities aboveground, and thus a limited impact on the soil-driven changes in leaf- and herbivore-associated microbial communities. In addition, we mainly focused on the herbivory-driven changes in microbiota composition in plant compartments, as they have immediate consequences for the host. Thus, the neoprene disks likely influenced the rhizosphere and root community composition, but this effect was separate from the influence of the rhizosphere on the herbivore and vice versa.

We showed that soil-driven changes in plant and herbivore microbiota composition occur via plants and not via microbial spillover between compartments. On the other hand, we found that microbial spillover from herbivores influenced the root and rhizosphere microbiota composition. Our results contribute to our understanding of the assembly of plant microbiota compartments and their responses to external factors. This is of high priority to enable the manipulation of plant microbiota. Given soil microbial communities have a strong effect on plant microbiota we might be able to use soil microbiota to enhance specific microbial functions or plant traits; for example, by steering soil microbial communities to negatively influence insect pests[23,24]. Thus, the management of the soil microbiota has the potential to promote food security and safety, restore damaged environments, and preserve endangered ecosystems.

## Methods
### Experimental design
We tested our hypothesis using a full factorial design, growing tomato plants (*Solanum lycopersicum* L. variety Moneymaker, Urban Farmer, Indianapolis, IN, USA) in microcosms containing three different soil microbial communities. We used three different soil inocula obtained from fields with different levels of disturbance (an agricultural field, field margin, and prairie) in order to ensure plants and herbivores were exposed to three soil communities known to vary in composition. This approach allowed us to tease apart contributions due to our treatments versus the original diversity of each inocula. To test whether the soil-driven effect on leaves and herbivore microbiota is generated by the plant or by the spillover of microorganisms from soil, the soil surface of half of the plants was covered with a black neoprene disk (~3 mm thick, Foam Factory Inc., MI, USA, Fig. 1), while the other half was left uncovered. Within each group, we exposed half of the plants to herbivory by *Manduca sexta* while the other half served as a control. We purchased eggs of *M. sexta* from the Great Lakes Hornworm (Romeo, MI, USA), placed them on an artificial diet (from the same provider) at room temperature, and waited until they reached the 2nd instar larva before inoculation. Each combination of soil inoculum ($n = 3$), coverage ($n = 2$), and herbivory ($n = 2$) was replicated eight times for a total of 96 plants.

### Microcosm setup
The soil to be used as inoculum was collected in June 2019 from three adjacent but differently managed grasslands at the Marion Campus of The Ohio State University (40.574 N, 83.088 W, Marion, OH, USA). We

https://doi.org/10.1038/s42003-024-06259-6                                                                                      **Article**

sampled soils with different levels of disturbance: an agricultural soil (collected in a field sown with soybean and subjected to corn-soybean rotation), a field margin (uncultivated area at the border between the prairie and the agricultural field), and a prairie (restored prairie left undisturbed for the past ~45 years).

Soil was collected from 0 to 15 cm of depth, sieved to 3 cm to remove large debris, homogenized, and stored at 4 °C. Sterilized background soil was generated from a mix of all three soils used for inoculation (1:1:1), which were then mixed with two parts sand (1 part combined soil: 2 parts sand). This mixture was sterilized by autoclaving at 121 °C for 3 h, allowing it to cool for 24 h, and autoclaved at 121 °C for a further 3 h. Sterile background soil was used to guarantee homogeneous soil conditions across the microcosms, which varied only in the composition of the soil inoculum.

Seeds were germinated on sterilized coir for 2 weeks in a greenhouse (average temperature 25 °C and photoperiod of 16 h light and 8 h dark). The microcosms were set up in 600 mL experimental deepots (Stewe & Sons Inc., Tangent, OR, USA; Fig. 1a, b). At the bottom of each pot, we added 100 mL of sterilized background soil followed by a 400 mL mix containing a soil inoculum mixture of soil from the field (180 mL, 60 mL of each of the three soils, two sterilized one alive) and sterilized background soil (220 mL). This approach controlled for soil physicochemical characteristics as in each pot two of the three inocula soils were autoclaved (e.g., the prairie soil treatment contained 60 mL of live soil from the prairie mixed with 60 mL of autoclaved field margin soil and 60 mL of autoclaved agricultural soil). Finally, 100 mL sterile background soil was added to the top of the pots to prevent contamination due to water splashing between pots. Thus, each pot contained 10% live soil. A single tomato seedling was transplanted to each pot. If the pot was assigned to the "covered" group, the soil surface was covered with a black neoprene disk (Fig. 1b). Plants were then randomized into two blocks and left to grow in an insect-screened greenhouse at an average temperature of 25 °C and a photoperiod of 16 h of light and 8 h of darkness. Plants were watered from the top with ~100 mL of tap water three times per week throughout the experiment.

Five weeks after the experimental setup, plants assigned to the herbivory treatment ($n = 48$) were exposed to herbivory by a single 2nd instar larva of *M. sexta*. All plants were screened using a microperforated plastic bag that allowed transpiration while preventing the escape of larvae. After 1 week, the larvae were collected, flash-frozen in liquid nitrogen, and stored at −80 °C. The microperforated plastic bags were removed, and the plants were left to grow for another week. From each pot, we collected three punch-holes from randomly selected leaves before placing them in a drying oven at 60 °C for 1 week. The roots were cleared from the loose surrounding soil, and ~25 mg of rhizosphere soil was collected from each plant by vigorously shaking the roots. The roots were then carefully washed, and after collecting ~25 mg of roots they were dried for 1 week at 60 °C. All samples for DNA extraction were immediately flash-frozen in liquid nitrogen, and after 1 week roots and shoots were weighed for dry biomass. Larvae were individually dissected to remove the intestine, which was transferred to a 2 mL tube and stored at −80 °C before DNA extraction, and carcasses were transferred to pre-weighted 2 mL tubes and dried in an oven for 1 week before being weighed.

### DNA extraction, library preparation, and sequencing
Each sample was lysed in extraction buffer using a bead-mill homogenizer, and total DNA was extracted using a phenol-chloroform protocol. In the case of roots and leaves, we did not surface sterilize samples, so we characterized both endophyte and epiphyte communities. After quality check, we prepared libraries targeting the bacterial 16 S rRNA gene (region V3-V4) using the primer pair 515 f/806rB[32] (~350 bp amplicon size). Amplifications were also carried out on DNA extracted from the soil inoculum and non-template controls, where the sample was replaced with nuclease-free water to account for possible contamination of instruments, reagents, and consumables used for DNA extraction. After this first PCR, samples were purified (Agencourt AMPure XP kit, Beckman Coulter) and used for a second short-run PCR to ligate Illumina adaptors. Libraries were then

purified again, quantified using a Qubit spectrophotometer (Thermo Fisher Scientific Inc.), normalized using nuclease-free water, pooled together, and sequenced on an Illumina NovaSeq 6000 SP 250PE flow cell at the Genomic Sciences Laboratory of North Carolina State University (Raleigh, NC, USA).

### Raw reads processing
Paired-end reads were processed using cutadapt[33] and DADA2 v1.22[34] implemented in the nf-core/ampliseq pipeline[35–37] to remove low-quality data, identify ASVs, and remove chimeras. Taxonomy was assigned using SILVA v138 database[38]. ASV sequences were aligned using MUSCLE[39] and a phylogenetic tree was generated using FastTree[40]. Data were processed and analyzed using R v4.1.2[41]. The ASV table, taxonomic information, metadata, and phylogenetic tree were then merged into a single object using *phyloseq*[42].

### Statistics and reproducibility
Phylogenetic diversity (Faith's index) was estimated for each sample using the package *picante*[43], and tests were performed for each compartment by fitting a linear-mixed effect model using the package *lme4*[44], soil inoculum (agricultural, margin, prairie), coverage (covered and control), herbivory (present and absent), and their interactions as fixed factors, and "block" as a random effect. The package *emmeans*[45] was used to infer pairwise contrasts (corrected using the false discovery rate, FDR), and the package *MuMIn* was used to estimate the $R^2$ values (https://CRAN.R-project.org/package=MuMIn) individually for each factor.

We tested the influence of the same factors (i.e., soil inoculum, coverage, and herbivory) on the structure of bacterial microbiomes in each compartment of our system using a multivariate approach. Distances between pairs of samples in terms of community composition were calculated using an unweighted UniFrac matrix and then visualized using an NMDS procedure. Differences between sample groups in the multivariate structure of their communities were inferred using permutational multivariate analysis of variance (PERMANOVA, 999 permutations), specifying compartment, soil inoculum, coverage, herbivory, and their interactions as fixed factors. Pairwise contrasts were inferred using the package *RVAideMemoire* (https://CRAN.R-project.org/package=RVAideMemoire), correcting p-values for multiple comparisons (FDR). In addition, we used the package *picante*[43] to calculate the beta Mean Nearest Taxon Distances (MNTD) to further test the influence of soil inoculum, coverage, and herbivory on the microbiome structure in each compartment. An MNTD is conceptually related to the phylogenetic diversity index, and informs us about the relatedness of pairs of species within a community. The output of MNTD produces a single value that can be analyzed by univariate analyses which we did by fitting the MNTD output to linear mixed-effect models specifying soil inoculum (agricultural, margin, prairie), coverage (covered and control), herbivory (present and absent), and their interactions as fixed factors, and block as a random effect. The package *emmeans* was used to infer pairwise contrasts (corrected using the false discovery rate, FDR).

The package *ComplexUpset* (https://CRAN.R-project.org/package=ComplexUpset) was used to visualize the number of ASVs unique to a compartment or shared between them. A chi-squared test was used to infer differences in the number of ASVs for each bin (i.e., a single compartment or intersection across two or more compartments).

The taxonomic composition was investigated by first normalizing the ASV table to account for sequencing bias using *DESeq2*. The same package was used to identify differences in the relative abundance of ASVs between the treatment groups (soil inoculum, cover, and herbivory) within each compartment.

Structural equation modelling was performed using the package *piecewiseSEM*[46]. We build the model testing the influence of our three factors (herbivory, soil inoculum, and cover) on the microbial community at each compartment (rhizosphere, roots, leaves, herbivore), and the effects of all factors and microbiota at each compartment on root, shoot, and herbivore biomass. For each compartment we performed a NMDS as above, and we used this as a proxy for the structure of microbial communities. Each model

within the structural equation model was a linear-mixed effects model that included block as a random effect.

## Data availability
Raw data is available at NCBI SRA under the Bioproject PRJNA910821.

## Code availability
The code to replicate analyses is available on Zenodo[47] and at: https://github.com/amalacrino/malacrino_and_bennett_CommsBio.

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

## Acknowledgements

AM is supported by the Italian Ministry of University and Research (MUR) through the PRIN 2022 PNRR program (project P2022KY74N, financed by the European Union - NextGenerationEU).

## Author contributions
Conceptualization: A.M., A.E.B.; Methodology: A.M., A.E.B.; Investigation: A.M.; Visualization: A.M.; Writing—original draft: A.M.; Writing—review, and editing: A.M., A.E.B.

## Competing interests
The authors declare no competing interests.
