## [Peer Review File · Communications Biology]

Reviewers' comments:

Reviewer #1 (Remarks to the Author):

Dear authors, Dear editors,

I have finished my assessment for 'Soil microbiota and herbivory drive the assembly of plant-associated microbial communities through different mechanisms' by Malacrinò and Bennett.

In this study, the authors set out to test the existence of microbial transfer between ecological compartments; soil, rhizosphere, root, shoot, and insect (chewing) herbivore in a tomato-hornworm system. In particular, they test the hypothesis that transfer of microbes is mediated via the host plant, and not by external spillover events. To do so, the authors sourced soils to grow their plants in from very different soil (microbial) environments, and use a neoprene disk treatment, which serves two purposes; soil/water splatter from the soil to the aboveground compartments, and the spillover of insect derived matter (frass, exuviae) to the soil.

The study is well-described, and has a number of interesting analyses and approaches, including the upset plots in Figure 3 and the SEM modelling approaches presented in Figure 6, which give a nice overview of the paths and effects observed in the study.

However, I do have some concerns about the approach that the authors could perhaps elaborate on, or perhaps clarify in the paper to help justify some of the methodological approaches.

General comments:

1) The authors chose neoprene disks to prevent up- and downward spillover events. On the one hand, I appreciate the material choice, as it is minimally damaging to the plant. On the other hand, I am a bit concerned by the fact that this neoprene disk is hard to 'control' for in the used study setting. Yes, the disk prevents spillover effectively. However, it may also cause significant changes in soil humidity, light environment, temperature, CO₂ build-up - to name a few - that simply are not present in the environment without the disk. It is hard to think of an appropriate control for this, but it might be one where a disk is present, but removed for watering, and where frass is collected on the disk and transferred to the soil regularly. Alternatively, the disks would be placed high enough not to cause any stagnant air? (I am just thinking out loud there). The reason I am somewhat concerned about this, is that in Figure 3, the huge differences between cover and no-cover treatment largely occur in rhizosphere- and root-involved compartments, which almost suggests that the neoprene disk conditions became an unintentional soil treatment. (For instance, if anything, I would have expected the cover to reduce # of ASVs, but it is substantially higher in roots). Have the authors tested how the neoprene disk affects microclimatic conditions at all, or considered the validity of their control (perhaps in earlier studies?).

2) I understand why the authors chose different soil inocula. For the purpose of testing microbial transfer, it is obviously convenient if the inocula are very different. However, it does come out of the blue a little bit. There was no clear introduction (or hypothesis) for the use of, or effects of, these different soils. Given that there is a gradient from natural to semi-disturbed to regularly disturbed soils, it might be interesting to formulate hypotheses around this? It almost feels like 'intermediate-disturbance', but it

was not framed that way. There must be a reason for including these three soil origins, and I think it would strengthen the paper to include this in the storyline.

3) I liked the SEM approach, but given the very brief description of how it was built (and I believe they are essentially two separate models, from what I gather?), it is hard for me to fully judge it/them. Did the final presented model in Figure 6 represent all hypothesized paths, or only paths that were significant? Some arrows are not present, but it is not clear whether they were tested/hypothesized. Due to my own research bias, I would expect an arrow from soil inoculum to insect biomass, for instance (but this is only one example). I think it would be good to have a figure that represents all the tested hypothetical paths, and then the significant paths in the final model. Using arrow size and color to represent strength and directions would make the figure more intuitive, possibly. Lastly, if it is true that two models were run, is it valid to include them in one figure? I would consider two separate panels for this perhaps.

Specific comments:

L11: remove comma after biology

L61: there is partial redundancy with the previous paragraph. This paragraph is better in my view. I think storywise, it would make sense to start early with the description of potential pathways, and then describe the evidence for it, and which gaps remain.

L70: It jumps quite oddly to a new subject here. Perhaps good to start a new paragraph?

L79: Explicitly naming the knowledge gap would be helpful.

L81: Perhaps it would be good to mention how herbivory was manipulated (absence presence, density, etc.)

L107: Rationale for this in the intro would be great, see my earlier comment.

L114: It is hard to say what the ratio of sterilized:live soil was, because the method is quite complex. For ease of understanding, I would mention the ratio early, before explaining the details. I would also opt for 'sterilized' soil, instead of 'sterile' soil. Nitpicky comment.

L122: It would be helpful to mention that two were sterilized and one was live?

L123: so this sterilized background was a mix of all three 1:1:1?

L125: just out of curiosity, as there are so many approaches to this. Why did you use this complex layering, and not just homogenize throughout?

L129: It reads as if the neoprene disk was only there to avoid damage. I think the function was different?

L133: Fig 1 is not super helpful in its current form. Perhaps an experimental design, which could include the soil layering/mixing approach, AND including the picture, would be more helpful.

L259: There is no Fig 2D?

L267: Not all figures are discussed chronologically (but this is easy to solve)

L400: Given the large number of significant differences in Figure 3, I don't think this statement is completely justified. Cover may not have changed much in terms of plant or insect performance, but it is clear that it did have a pretty huge effect on richness in microbiota. I think this should at least be given some attention in the discussion.

I hope that some of my suggestions may help the authors to further improve their work, and I wish them all the best in revising their manuscript.

Sincerely,

Robin Heinen, Technical University of Munich

Reviewer #2 (Remarks to the Author):

This manuscript investigates the contribution of different drivers (herbivory, soil cover and soil inoculum) on the microbial assemblages in different compartments of tomato plants (rhizosphere, roots, leaves), to elucidate whether the soil-driven effect that has been consistently reported in previous reports is mediated by the plant itself or my microbial spillover. The study provides novel insights to answer a relevant ecological question and uses an experimental approach to test their hypotheses. I really enjoyed reading the manuscript and I believe that it is a significant contribution to the field.

My main comment would be that some aspects of the experimental design seem to have been disregarded, maybe because they were not essential to answer the specific question that the authors asked. This is a pity, as more data could have been gathered from this experiment. For example, why not including a control with sterile inoculum only? It would have been interesting, to test the efficiency of soil sterilization. Also, why not consider the initial soil inoculum in the amplicon sequencing? If soil-driven effects are investigated, microbial spillover from soil is more likely to occur from the bulk soil than from the rhizosphere. Similarly, a herbivore effect on the soil microbiota (through frass for example) is more likely to be observed on the bulk soil and not on the rhizosphere. I understand that the focus of this work that the plant-associated microbiota but this somewhat should be discussed, and the choice of not including soil microbiota in the equation should be justified.

Minor comments are enlisted below.

Keywords should not repeat words already mentioned in the title. Maybe some more informative keywords could be used than 16S rRNA or metabarcoding?

Introduction

L32: what do you mean by “directly or indirectly colonizing different plant compartments”? A few examples on direct or indirect mechanisms could help clarify this statement.

Methods

L111: Grassland or soybean field? It may be my misconception but by “differently managed grasslands” (L106) I understand that land use is similar but that management intensity may differ. I suggest stating since the beginning that land use is different (prairie, field margin and soybean field, as you mention later in the text).

L138: all plants were covered with a microperforated plastic bag. Do you refer to plants assigned to the herbivory treatment or to all plants? If only those assigned to the herbivory treatment were covered, is it possible that the plastic bag may have influenced the microbiota leaf colonization by preventing air-borne microbes to establish on the leaf surface?

L152: It is unclear in the Methods description if “roots” refer to endophytic microbiota or rhizoplane. If the endophytic compartment is the one being considered, the disinfection procedure should be

described in more detail. Same comment for the leaves: phyllosphere or endophytic microbiota? How did you recover the leaf microbes for DNA extraction?

Is it possible that the soil coverage may have influence soil temperature or humidity and thus have an indirect effect on the soil microbiota?

Results

Fig. 2: please keep the same order for the treatments as the one mentioned in the Methods (prairie, field margin, soybean field), just to avoid any confusion.

Reviewer #3 (Remarks to the Author):

In their study “Soil microbiota and herbivory drive the assembly of plant associated microbial communities through different mechanisms”, Malacrino and Bennet investigate whether herbivory and the soil microbial community influence each other via the plant or directly via spillover effects. To this end, they conducted an experiment using three different soils (aka bacterial communities) in which tomato plants were grown with and without herbivory and with and without a soil cover to prevent direct spillover. The bacterial communities were assessed in the herbivores, the plant leaves, the roots and the rhizosphere. The results suggest that there is an effect of the soil bacterial community on the herbivore bacterial community that is mediated by the plant. The root and rhizosphere bacterial community are also influenced by herbivory, but this effect is likely both mediated via the plant and derived from direct spillover effects.

The study was conducted rigorously taking into account possible confounding factors. The appropriate statistical methods were used for analysis and the manuscript is written clearly and concisely.

Most importantly, this study answers the question on how the soil, plant, and herbivore bacterial community influence each other, which has been the object of much speculation.

Most comments below are simply for clarification and to improve readability.

General comments

1. Since in this study the bacterial, but not the fungal community composition were assessed I suggest to use “bacterial community” instead of “microbial community”.

Introduction

2. In the first paragraph it should be stated more clearly which plant compartments the authors are talking about. That would e.g. make it clearer how herbivores could indirectly affect the plant microbiome.

3. L. 32: Do you mean the plant endophytic microbiome?

4. L. 50: Are those the same compartments as talked about in the first paragraph?

5. L. 52: Please define spillover.

6. L. 64: A very similar sentence can be found in L. 32 and following. Consider changing this part to make

it less repetitive.

7. L. 75: This thought was also already mentioned above in L. 38, 39. Maybe you could shorten the first paragraph by removing these specific sentences and then let them come back in the third paragraph.

Methods

8. It does not become completely clear why both permanovas and MNTDs were calculated. I get the impression that one of those tests would have been sufficient.

9. L. 126: I suggest to define the abbreviation SBS already in line 113 when "sterile background soil" is first used.

Results

10. L. 259: I could not find figure 2D.

Discussion

11. I miss the discussion on differential abundances. Since they are not essential to the conclusions it might be possible to leave them out, but if they stay in the results they should also be discussed.

12. L. 437 and in the results: It might be that I am not an expert in modelling, but I wonder how soil inoculum can negatively influence insect biomass. What was the control for this? Soil without an inoculum?

Kind regards,

Viola Kurm

Reviewer #1

Comment #1. I have finished my assessment for 'Soil microbiota and herbivory drive the assembly of plant-associated microbial communities through different mechanisms' by Malacrinò and Bennett. In this study, the authors set out to test the existence of microbial transfer between ecological compartments; soil, rhizosphere, root, shoot, and insect (chewing) herbivore in a tomato-hornworm system. In particular, they test the hypothesis that transfer of microbes is mediated via the host plant, and not by external spillover events. To do so, the authors sourced soils to grow their plants in from very different soil (microbial) environments, and use a neoprene disk treatment, which serves two purposes; soil/water splatter from the soil to the aboveground compartments, and the spillover of insect derived matter (frass, exuviae) to the soil. The study is well-described, and has a number of interesting analyses and approaches, including the upset plots in Figure 3 and the SEM modelling approaches presented in Figure 6, which give a nice overview of the paths and effects observed in the study. However, I do have some concerns about the approach that the authors could perhaps elaborate on, or perhaps clarify in the paper to help justify some of the methodological approaches.

→ Response. Thanks for the positive feedback. Please see our responses below.

Comment #2. The authors chose neoprene disks to prevent up- and downward spillover events. On the one hand, I appreciate the material choice, as it is minimally damaging to the plant. On the other hand, I am a bit concerned by the fact that this neoprene disk is hard to 'control' for in the used study setting. Yes, the disk prevents spillover effectively. However, it may also cause significant changes in soil humidity, light environment, temperature, CO₂ build-up - to name a few - that simply are not present in the environment without the disk. It is hard to think of an appropriate control for this, but it might be one where a disk is present, but removed for watering, and where frass is collected on the disk and transferred to the soil regularly. Alternatively, the disks would be placed high enough not to cause any stagnant air? (I am just thinking out loud there). The reason I am somewhat concerned about this, is that in Figure 3, the huge differences between cover and no-cover treatment largely occur in rhizosphere- and root-involved compartments, which almost suggests that the neoprene disk conditions became an unintentional soil treatment. (For instance, if anything, I would have expected the cover to reduce # of ASVs, but it is substantially higher in roots). Have the authors tested how the neoprene disk affects microclimatic conditions at all, or considered the validity of their control (perhaps in earlier studies?).

→ Response. Thanks for your comment. We agree with your comments, and we now discuss the impact of the neoprene disks on rhizosphere and root microbiome composition in more detail in the discussion of the manuscript (L444-454):

[...] While covering soil with a neoprene disk allowed us to separate aboveground and belowground compartments from microbial spillover, it might also represent a caveat of our study. In particular, the black neoprene disks might have influenced the temperatures and/or humidity due to the black cover, similarly to black plastic mulch which increases soil temperatures and plant biomass. On the other hand, the soil cover likely had a limited role in influencing the microbial communities aboveground, and thus a limited impact on the soil-driven changes in leaf- and herbivore-associated microbial communities. In addition, we mainly focused on the herbivory-driven changes in microbiota composition in plant compartments, as they have immediate

consequences for the host. Thus, the neoprene disks likely influenced the rhizosphere and root community composition, but this effect was separate from the influence of the rhizosphere on the herbivore and vice versa [...]

Comment #3. I understand why the authors chose different soil inocula. For the purpose of testing microbial transfer, it is obviously convenient if the inocula are very different. However, it does come out of the blue a little bit. There was no clear introduction (or hypothesis) for the use of, or effects of, these different soils. Given that there is a gradient from natural to semi-disturbed to regularly disturbed soils, it might be interesting to formulate hypotheses around this? It almost feels like 'intermediate-disturbance', but it was not framed that way. There must be a reason for including these three soil origins, and I think it would strengthen the paper to include this in the storyline.

→ Response. Thanks for your comment. We agree with you that the level of disturbance is an interesting factor to take into account. However, we selected those inocula with the sole purpose of having three different starting communities. We feel that we do not have the correct design and level of replication to be able to ask questions on how the level of soil disturbance influence the assembly of plant and herbivore microbiome. Our microcosm setup provides disturbance to the soil, and this would not allow us to accurately disentangle the effect of disturbance in the field from the one generated during the experiment. The question you pose is very interesting, and with the correct setup it would be very interesting to test.

Comment #4. I liked the SEM approach, but given the very brief description of how it was built (and I believe they are essentially two separate models, from what I gather?), it is hard for me to fully judge it/them. Did the final presented model in Figure 6 represent all hypothesized paths, or only paths that were significant? Some arrows are not present, but it is not clear whether they were tested/hypothesized. Due to my own research bias, I would expect an arrow from soil inoculum to insect biomass, for instance (but this is only one example). I think it would be good to have a figure that represents all the tested hypothetical paths, and then the significant paths in the final model. Using arrow size and color to represent strength and directions would make the figure more intuitive, possibly. Lastly, if it is true that two models were run, is it valid to include them in one figure? I would consider two separate panels for this perhaps.

→ Response. Thanks for your comment. Based on comments from another reviewer we revised the analysis approach we used for SEM, and now we use a piecewise SEM to have more control on our hypotheses. We are now able to build a single model, and we report the tested but not significant paths in the figure as grey dashed lines.

Comment #5. L11: remove comma after biology

→ Response. Done.

Comment #6. L61: there is partial redundancy with the previous paragraph. This paragraph is better in my view. I think storywise, it would make sense to start early with the description of potential pathways, and then describe the evidence for it, and which gaps remain.

→ Response. Thanks for your comment. We changed the order of those paragraphs and merged them.

Comment #7. L70: It jumps quite oddly to a new subject here. Perhaps good to start a new paragraph?

→ Response. Thanks for your comment. We rephrased this bit to make the text flow better.

Comment #8. L79: Explicitly naming the knowledge gap would be helpful.

→ Response. Thanks for your comment. Following your guidance above we rewrote this paragraph to make the knowledge gap more apparent (L76):

[...] In this study, we aimed to clarify the mechanisms underlying the reciprocal influence of microbiomes in a plant-herbivore system [...]

Comment #9. L81: Perhaps it would be good to mention how herbivory was manipulated (absence presence, density, etc.)

→ Response. Done.

Comment #10. L107: Rationale for this in the intro would be great, see my earlier comment.

→ Response. Please see our response to your comment #3.

Comment #11. L114: It is hard to say what the ratio of sterilized:live soil was, because the method is quite complex. For ease of understanding, I would mention the ratio early, before explaining the details. I would also opt for 'sterilized' soil, instead of 'sterile' soil. Nitpicky comment.

→ Response. Thanks for your comment. We revised the text of the following paragraph according to your comment (L130):

[...] Thus, each pot contained 10% live soil [...]

Comment #12. L122: It would be helpful to mention that two were sterilized and one was live?

→ Response. Done (L124):

*[...] 400 mL mix containing a soil inoculum mixture of soil from the field (180 mL, 60 mL of each of the three soils, **two sterilized one alive**) and sterilized background soil (220 mL) [...]*

Comment #13. L123: so this sterilized background was a mix of all three 1:1:1?

→ Response. Yes, we now clarify this in the text (L124):

*[...] 400 mL mix containing a soil inoculum mixture of soil from the field (180 mL, **60 mL of each of the three soils, two sterilized one alive**) and sterilized background soil (220 mL) [...]*

Comment #14. L125: just out of curiosity, as there are so many approaches to this. Why did you use this complex layering, and not just homogenize throughout?

→ Response. Thank you for your interest in our approach. Layering sterile soil on the top and bottom prevents contamination, especially on the top due to potential splashing between pots when watering. We now acknowledge this in the text (L128):

[...] 100 mL sterile background soil was added to the top of the pots to prevent contamination due to water splashing between pots [...]

Comment #15. L129: It reads as if the neoprene disk was only there to avoid damage. I think the function was different?

→ Response. Thanks, we removed the confusing text.

Comment #16. L133: Fig 1 is not super helpful in its current form. Perhaps an experimental design, which could include the soil layering/mixing approach, AND including the picture, would be more helpful.

→ Response. Thanks for your comment. We now include an additional panel in Figure 1 showing how pots were assembled.

Comment #17. L259: There is no Fig 2D?

→ Response. Fixed. We actually meant 2C.

Comment #18. L267: Not all figures are discussed chronologically (but this is easy to solve)

→ Response. Thanks for your comment. We think you are referring to Fig. 2, where panel C was mentioned before panel B. This is now fixed by adjusting the order of the panels within the figure.

Comment #19. L400: Given the large number of significant differences in Figure 3, I don't think this statement is completely justified. Cover may not have changed much in terms of plant or insect performance, but it is clear that it did have a pretty huge effect on richness in microbiota. I think this should at least be given some attention in the discussion.

→ Response. To acknowledge your comment, we have rephrased the sentence in line 400 (now L404) to state "*little influence*" instead of "*no influence*". Fig. 3 shows that the number of ASVs shared between compartment was influenced by cover. However, all other tests (phylogenetic diversity, MNTD, PERMANOVA, differential abundance) show a limited impact of soil cover on the microbial community in each compartment. This suggests that soil cover played a role in modulating the spillover of microbes between aboveground and belowground compartments, as we expected, but the cover itself had little overall influence on the structure of the microbial communities in different compartments.

Comment #20. I hope that some of my suggestions may help the authors to further improve their work, and I wish them all the best in revising their manuscript.

→ Response. Thank you!

Reviewer #2

Comment #1. This manuscript investigates the contribution of different drivers (herbivory, soil cover and soil inoculum) on the microbial assemblages in different compartments of tomato plants (rhizosphere, roots, leaves), to elucidate whether the soil-driven effect that has been consistently reported in previous reports is mediated by the plant itself or my microbial spillover. The study provides novel insights to answer a relevant ecological question and uses an experimental approach to test their hypotheses. I really enjoyed reading the manuscript and I believe that it is a significant contribution to the field.

→ Response. Thanks for your positive feedback!

Comment #2. My main comment would be that some aspects of the experimental design seem to have been disregarded, maybe because they were not essential to answer the specific question that the authors asked. This is a pity, as more data could have been gathered from this experiment. For example, why not including a control with sterile inoculum only? It would have been interesting, to test the efficiency of soil sterilization.

→ Response. Thanks for your comments. Unfortunately, sterilizing soil inocula does not destroy DNA which can be still detected even after organisms have been killed. Thus, sequencing sterile inocula would not have tested the efficiency of soil sterilization.

Comment #3. Also, why not consider the initial soil inoculum in the amplicon sequencing? If soil-driven effects are investigated, microbial spillover from soil is more likely to occur from the bulk soil than from the rhizosphere.

→ Response. We struggled to understand what this comment was requesting. We think the reviewer is enquiring why we did not compare the starting and finishing soil inocula communities? If so, to statistically robustly compare these communities would have required pairing samples of the initial inocula with each replicate which would have been unfeasible as it would have doubled our sequencing costs. Or is the reviewer enquiring why we did not examine both the rhizosphere and bulk soils? Bulk soils have significantly lower diversity than rhizosphere soils and are a subset of those soils, thus by sampling rhizosphere soils we are also sampling bulk soils.

Comment #4. Similarly, a herbivore effect on the soil microbiota (through frass for example) is more likely to be observed on the bulk soil and not on the rhizosphere. I understand that the focus of this work that the plant-associated microbiota but this somewhat should be discussed, and the choice of not including soil microbiota in the equation should be justified.

→ Response. Please see our response to your comment #3. Bulk soils have significantly lower diversity than rhizosphere soils and are a subset of those soils, thus by sampling rhizosphere soils we are also sampling bulk soils.

Comment #5. Keywords should not repeat words already mentioned in the title. Maybe some more informative keywords could be used than 16S rRNA or metabarcoding?

→ Response. We revised the list of keywords:

Keywords: *plant microbiota; plant-microbe-insect interactions; microcosm; amplicon metagenomics*

Comment #6. L32: what do you mean by “directly or indirectly colonizing different plant compartments”? A few examples on direct or indirect mechanisms could help clarify this statement.

→ Response. Thanks for your comment. We removed “directly or indirectly” and added “(e.g., *rhizosphere, roots, leaves*), both as *endophytes and epiphytes*” in L34 to help clarify this statement.

Comment #7. L111: Grassland or soybean field? It may be my misconception but by “differently managed grasslands” (L106) I understand that land use is similar but that management intensity may differ. I suggest stating since the beginning that land use is different (prairie, field margin and soybean field, as you mention later in the text).

→ Response. Thanks for your comment. Based on all reviewer’s comments we revised L92 to remove an emphasis on land use type and discuss disturbance instead.

The section now reads:

[...] We used three different soil inocula obtained from fields with different levels of disturbance (an agricultural field, field margin, and prairie) in order to ensure plants and herbivores were exposed to three soil communities known to vary in composition. This approach allowed us to tease apart contributions due to our treatments versus the original diversity of each inocula [...]

We also altered L109 to read:

[...] We sampled soils with different levels of disturbance: an agricultural soil (collected in a field sown with soybean and subjected to corn-soybean rotation), a field margin (uncultivated area at the border between the prairie and the agricultural field), and a prairie (restored prairie left undisturbed for the past ~45 years) [...]

Comment #8. L138: all plants were covered with a microperforated plastic bag. Do you refer to plants assigned to the herbivory treatment or to all plants? If only those assigned to the herbivory treatment were covered, is it possible that the plastic bag may have influenced the microbiota leaf colonization by preventing air-borne microbes to establish on the leaf surface?

→ Response. Thanks for your comment. We indeed mean that all plants (with or without herbivores) were covered.

Comment #9. L152: It is unclear in the Methods description if “roots” refer to endophytic microbiota or rhizoplane. If the endophytic compartment is the one being considered, the disinfection procedure should be described in more detail. Same comment for the leaves: phyllosphere or endophytic microbiota? How did you recover the leaf microbes for DNA extraction?

→ Response. Thanks for your comment. We now clarify (L158):

[...] In the case of roots and leaves, we did not surface sterilize samples, so we characterized both endophyte and epiphyte communities [...]

Comment #10. Is it possible that the soil coverage may have influence soil temperature or humidity and thus have an indirect effect on the soil microbiota?

→ Response. Thanks for your comment. We now discuss this aspect (L444-454):

[...] While covering soil with a neoprene disk allowed us to separate aboveground and belowground compartments from microbial spillover, it might also represent a caveat of our study. In particular, the black neoprene disks might have influenced the temperatures and/or humidity due to the black cover, similarly to black plastic mulch which increases soil temperatures and plant biomass. On the other hand, the soil cover likely had a limited role in influencing the microbial communities aboveground, and thus a limited impact on the soil-driven changes in leaf- and herbivore-associated microbial communities. In addition, we mainly focused on the herbivory-driven changes in microbiota composition in plant compartments, as they have immediate

consequences for the host. Thus, the neoprene disks likely influenced the rhizosphere and root community composition, but this effect was separate from the influence of the rhizosphere on the herbivore and vice versa [...]

Comment #11. Fig. 2: please keep the same order for the treatments as the one mentioned in the Methods (prairie, field margin, soybean field), just to avoid any confusion.

→ Response. We rephrased the methods section to have the same order of all the figures comparing across soil inocula (L109):

*[...] We sampled soils with different levels of disturbance: an **agricultural soil** (collected in a field sown with soybean and subjected to corn-soybean rotation), a **field margin** (uncultivated area at the border between the prairie and the agricultural field), and a **prairie** (restored prairie left undisturbed for the past ~45 years) [...]*

Reviewer #3

Comment #1. In their study “Soil microbiota and herbivory drive the assembly of plant associated microbial communities through different mechanisms”, Malacrino and Bennet investigate whether herbivory and the soil microbial community influence each other via the plant or directly via spillover effects. To this end, they conducted an experiment using three different soils (aka bacterial communities) in which tomato plants were grown with and without herbivory and with and without a soil cover to prevent direct spillover. The bacterial communities were assessed in the herbivores, the plant leaves, the roots and the rhizosphere. The results suggest that there is an effect of the soil bacterial community on the herbivore bacterial community that is mediated by the plant. The root and rhizosphere bacterial community are also influenced by herbivory, but this effect is likely both mediated via the plant and derived from direct spillover effects. The study was conducted rigorously taking into account possible confounding factors. The appropriate statistical methods were used for analysis and the manuscript is written clearly and concisely. Most importantly, this study answers the question on how the soil, plant, and herbivore bacterial community influence each other, which has been the object of much speculation. Most comments below are simply for clarification and to improve readability.

→ Response. Thanks for your positive feedback!

Comment #2. Since in this study the bacterial, but not the fungal community composition were assessed I suggest to use “bacterial community” instead of “microbial community”.

→ Response. Done.

Comment #3. In the first paragraph it should be stated more clearly which plant compartments the authors are talking about. That would e.g. make it clearer how herbivores could indirectly affect the plant microbiome.

→ Response. We added “(e.g., rhizosphere, roots, leaves)” to L34.

Comment #4. L. 32: Do you mean the plant endophytic microbiome?

→ Response. Actually, both endophytes and epiphytes. We clarified this sentence by adding “both as endophytes and epiphytes” to L34.

Comment #5. L. 50: Are those the same compartments as talked about in the first paragraph?

→ Response. Yes. We now clarify this in the first paragraph.

Comment #6. L. 52: Please define spillover.

→ Response. Thanks for your comment. We add the definition of spillover and present examples for it (L58):

[...] In contrast to the 'via plant' hypothesis, microbiomes in a plant-herbivore system might influence each other by spillover (i.e., movement of microbes between compartments). For example, herbivores can influence belowground microbial communities through microorganisms associated with frass or honeydew. Similarly, soil microorganisms can reach aboveground compartments when irrigation water or rain splashes over the soil surface, and droplets containing microorganisms come into contact with leaves [...]

Comment #7. L. 64: A very similar sentence can be found in L. 32 and following. Consider changing this part to make it less repetitive.

→ Response. We have deleted the sentence in the introductory paragraph to avoid repetition with material in the second, now reorganized, paragraph.

Comment #8. L. 75: This thought was also already mentioned above in L. 38, 39. Maybe you could shorten the first paragraph by removing these specific sentences and then let them come back in the third paragraph.

→ Response. Based on guidance from Reviewer #1 (Comments 6 and 7) we have reorganized the second and third paragraphs in a manner that we feel addresses this comment and helps the text flow better.

Comment #9. It does not become completely clear why both permanovas and MNTDs were calculated. I get the impression that one of those tests would have been sufficient.

→ Response. Thanks for your comment. While MNTD and PERMANOVA might look similar, they tell different aspects of the story. An MNTD is conceptually more closely related to the phylogenetic diversity index (see <https://cran.r-project.org/web/packages/picante/vignettes/picante-intro.pdf>) and tells us how closely related are pairs of species within a community, and produces a single value that can be used for univariate analyses. On the other hand, PERMANOVA looks at the multivariate structure of the community, and allows us to test the effect of different factors on the multivariate composition of the microbial community. To clarify this, we have added similar text to L199:

[...] An MNTD is conceptually related to the phylogenetic diversity index, and informs us about the relatedness of pairs of species within a community. The output of MNTD produces a single value that can be analyzed using for univariate analyses which we did by fitting the MNTD output to linear mixed-effect models [...]

Comment #10. L. 126: I suggest to define the abbreviation SBS already in line 113 when "sterile background soil" is first used.

→ Response. We removed this abbreviation as it not further used in the manuscript.

Comment #11. L. 259: I could not find figure 2D.

→ Response. We actually meant Fig. 2C. This is now fixed.

Comment #. I miss the discussion on differential abundances. Since they are not essential to the conclusions it might be possible to leave them out, but if they stay in the results they should also be discussed.

→ Response. Thanks for your comment. We agree that focusing on discussing variation in single taxa is not essential for the conclusion, but we believe that this additional analysis supports our conclusion that soil is the major driver of variation in the plant microbiome in other ways mentioned in L372:

[...] In particular, the soil inoculum explained a wider portion of the variance in microbiota diversity and structure in roots and herbivore microbiota, while this effect was much lower in the rhizosphere and leaves. In addition, we found a higher number of ASVs significantly affected by soil inoculum in the roots and herbivores compared to the other compartments. These results are similar to those of our previous study, where we observed differences between high- and low-diversity microbial inocula in the microbiota of plants and herbivores [...]

Comment #. L. 437 and in the results: It might be that I am not an expert in modelling, but I wonder how soil inoculum can negatively influence insect biomass. What was the control for this? Soil without an inoculum?

→ Response. Thanks for your comment. Our results suggest that variation in soil inoculum drives variation in the herbivore-associated microbiome, and this variation has negative effects on the herbivore biomass. We expanded the discussion to clarify this result (L418):

[...] Interestingly, we observed that variation in soil inoculum negatively influenced insect biomass through changes in the plant and insect microbiota. This supports the idea that steering the soil microbiota may be an effective way to achieve sustainable pest management [...]

REVIEWERS' COMMENTS:

Reviewer #1 (Remarks to the Author):

Dear authors, Dear editor,

I have now read the revised version of the manuscript COMMSBIO-23-4291A, and I want to start off by complimenting the authors in the way they addressed comments, and structured the rebuttal letter, with thoughtful answers where needed, and nice inserts of changed texts where appropriate. I wish all authors would adopt such an approach to revising manuscripts, as it made it a breeze.

Continuing with the quality of the revisions, I am also generally very pleased with these. In my view the addition of the layering approach figure is immediately clear. I also really like the way the authors dealt with my SEM comments. The figure now is standalone and with the revised legend is clear and intuitive. Great.

I also very much appreciate the inclusion of the caveat; this is how I believe caveats should be acknowledged. No study is perfect, and I think we can learn most if we agree that future studies should try to avoid the same caveats when they can. The authors could potentially widen it to more than just warming of soils. I also would believe that the aboveground parts could certainly be affected by warming of soils, via plant-mediated pathways. I would be careful in making too strong of a statement there.

I agree with the authors that the inclusion of disturbance levels would not be of much interest in the current study design. Indeed, replication of disturbance level sites, or independent samplings from one site, at the least, would need to be present for this to be reliably tested. I stand corrected. Nevertheless, the same applies to Figure 2 then, which may need to be addressed in the discussion. Given that the study is mostly about the comparisons of compartments, not the variation in communities, I do not think this is particularly problematic in this case. I also want to add that I do appreciate the changes to Figure 2. Despite the design not being the most appropriate to draw firm conclusions, you could also speculate that the patterns seem to follow the gradient, and that this may be one of the reasons. I also think including the specific rationale that soils were selected to obtain the most different soil inocula, may be helpful to a reader (somewhere around L150 would work).

Other than these fairly minor textual suggestions, I have no further comments.

Best wishes,
Robin Heinen, Technical University of Munich.

Reviewer #2 (Remarks to the Author):

I am satisfied with the authors' answers to all comments made by the three reviewers, including myself.

My only remaining comment to the authors would be that bulk soil diversity is not necessarily lower than rhizosphere diversity (Answers to comments 3 and 4 by Reviewer 2, myself). See for example: Essel et al. 2019 (Soil Tillage Res), Liu et al. 2018 (Applied Soil Ecol), Xu et al. 2023 (Rhizosphere) or Chen et al. 2023 (Ecol. Indic.)

Rhizosphere communities are selected by the plant from the reservoir that is the bulk soil. So it would be relevant to address, or at least discuss, the frass effect or microbial spillover on the bulk soil and not only on the rhizosphere soil, where it is more likely to occur.

Reviewer #3 (Remarks to the Author):

I only have very few comments left.

L. 202: The sentence appears a bit scrambled. Perhaps you mean "...that can be analyzed by univariate analyses..."

Comment former L. 437, now L. 418

I think this point needs further clarification. I understand what you mean, but now it reads as if insect biomass lower the more variable the microbiome is, which is probably not what you want to say. It would actually be helpful to explain how the multilevel factor soil inoculum was incorporated into the SEM, since I understand that this is not a straightforward process.

Reviewer #1

Dear authors, Dear editor,

I have now read the revised version of the manuscript COMMSBIO-23-4291A, and I want to start off by complimenting the authors in the way they addressed comments, and structured the rebuttal letter, with thoughtful answers where needed, and nice inserts of changed texts where appropriate. I wish all authors would adopt such an approach to revising manuscripts, as it made it a breeze.

Continuing with the quality of the revisions, I am also generally very pleased with these. In my view the addition of the layering approach figure is immediately clear. I also really like the way the authors dealt with my SEM comments. The figure now is standalone and with the revised legend is clear and intuitive. Great.

I also very much appreciate the inclusion of the caveat; this is how I believe caveats should be acknowledged. No study is perfect, and I think we can learn most if we agree that future studies should try to avoid the same caveats when they can. The authors could potentially widen it to more than just warming of soils. I also would believe that the aboveground parts could certainly be affected by warming of soils, via plant-mediated pathways. I would be careful in making too strong of a statement there.

→ Response. Thank you for your positive feedback.

I agree with the authors that the inclusion of disturbance levels would not be of much interest in the current study design. Indeed, replication of disturbance level sites, or independent samplings from one site, at the least, would need to be present for this to be reliably tested. I stand corrected. Nevertheless, the same applies to Figure 2 then, which may need to be addressed in the discussion. Given that the study is mostly about the comparisons of compartments, not the variation in communities, I do not think this is particularly problematic in this case. I also want to add that I do appreciate the changes to Figure 2. Despite the design not being the most appropriate to draw firm conclusions, you could also speculate that the patterns seem to follow the gradient, and that this may be one of the reasons. I also think including the specific rationale that soils were selected to obtain the most different soil inocula, may be helpful to a reader (somewhere around L150 would work).

→ Response. We now include a speculation on the soil disturbance levels (L200).

[...] It is interesting to note that the changes driven by soil inoculum on the herbivore microbiome seem to follow a gradient of disturbance from most disturbed (agricultural soil) to the less disturbed (prairie soil), although further evidence is needed to test this hypothesis [...]

The rationale for soil selection is already reported at L311:

We sampled soils with different levels of disturbance: an agricultural soil (collected in a field sown with soybean and subjected to corn-soybean rotation), a field margin (uncultivated area at the border between the prairie and the agricultural field), and a prairie (restored prairie left undisturbed for the past ~45 years).

Other than these fairly minor textual suggestions, I have no further comments.

→ Response. Thank you for your positive feedback.

Reviewer #2

I am satisfied with the authors' answers to all comments made by the three reviewers, including myself.

→ Response. Thank you for your positive feedback.

My only remaining comment to the authors would be that bulk soil diversity is not necessarily lower than rhizosphere diversity (Answers to comments 3 and 4 by Reviewer 2, myself). See for example: Essel et al. 2019 (Soil Tillage Res), Liu et al. 2018 (Applied Soil Ecol), Xu et al. 2023 (Rhizosphere) or Chen et al. 2023 (Ecol. Indic.) Rhizosphere communities are selected by the plant from the reservoir that is the bulk soil. So it would be relevant to address, or at least discuss, the frass effect or microbial spillover on the bulk soil and not only on the rhizosphere soil, where it is more likely to occur.

→ Response. We acknowledge this caveat in the discussion (L263):

[...] While our study tests the effects of herbivory on the rhizosphere microbial community, it does not provide evidence on its effects on the microbiota of the bulk soil, and future studies might focus on investigating the consequences of herbivory-driven changes on the soil microbiome and their effect on the wider ecological community [...]

Reviewer #3

I only have very few comments left.

L. 202: The sentence appears a bit scrambled. Perhaps you mean "...that can be analyzed by univariate analyses..."

→ Response. Thanks for your feedback. We fixed this bit.

Comment former L. 437, now L. 418

I think this point needs further clarification. I understand what you mean, but now it reads as if insect biomass lower the more variable the microbiome is, which is probably not what you want to say. It would actually be helpful to explain how the multilevel factor soil inoculum was incorporated into the SEM, since I understand that this is not a straightforward process.

→ Response. Thanks for your feedback. Our SEM approach shows that variation in the soil inoculum has an effect on the structure of the herbivore microbiome. This, in turn, influences the insect biomass. This last effect is not driven by the identity of the soil inoculum, but by its structure represented as the NMDS1 axis as a proxy.